# The Hepcidin-25/Ferritin Ratio Is Increased in University Rugby Players with Lower Fat Mass

**DOI:** 10.3390/nu13092993

**Published:** 2021-08-27

**Authors:** Shinsuke Nirengi, Mami Fujibayashi, Sachiko Furuno, Shin Sukino, Akiko Suganuma, Yaeko Kawaguchi, Yasuharu Kawase, Kazuhiko Kotani, Naoki Sakane

**Affiliations:** 1Division of Preventive Medicine, Clinical Research Institute, National Hospital Organization Kyoto Medical Center, Kyoto 612-8555, Japan; shi.nirengi@gmail.com (S.N.); cocoro8888@gmail.com (S.S.); jjddooiitt11@yahoo.co.jp (A.S.); kawaguchiyaeko@yahoo.co.jp (Y.K.); kazukotani@jichi.ac.jp (K.K.); 2Dorothy M. Davis Heart and Lung Research Institute, Department of Physiology and Cell Biology, The Ohio State University Wexner Medical Center, Columbus, OH 43210, USA; 3Faculty of Agriculture, Department of Food Science and Human Nutrition, Setsunan University, Osaka 573-0101, Japan; mami.fujibayashi@setsunan.ac.jp (M.F.); sachiko.furuno@setsunan.ac.jp (S.F.); 4Division of Physical and Health Education, Setsunan University, Osaka 572-8508, Japan; kawase@mpg.setsunan.ac.jp; 5Center for Community Medicine, Division of Community and Family Medicine, Jichi Medical University, Tochigi 329-0498, Japan

**Keywords:** hepcidin-25, iron status, rugby football players

## Abstract

Hepcidin-25 is suggested as a surrogate iron status marker in athletes who show exercise-induced anemia; however, the implications of hepcidin concentration in this population remain poorly understood. This study aimed to investigate the relationship between hepcidin and body fat levels in rugby football players. We included 40 male university rugby football players (RUG) and 40 non-athlete controls. All participants underwent an anthropometric analysis and blood testing that included both hepcidin-25 and ferritin levels. The hepcidin-25 level was slightly (11.6%, *p* = 0.50) higher, and the ferritin level was significantly (35.9%, *p* < 0.05) lower, in the RUG group than in controls. The hepcidin-25 to-ferritin ratio was significantly higher (62.5%, *p* < 0.05) in the RUG group. While significant U-shaped correlations were observed between the body fat and ferritin levels in both groups, the correlations between the hepcidin levels and fat mass index were significantly higher in the RUG group (RUG: *r* = 0.79, controls: *r* = 0.45). Notably, the RUG with the lower fat mass index group had a higher hepcidin-25 level, lower ferritin level, and then significantly higher hepcidin-25/ferritin ratio. The hepcidin-25/ferritin ratio may serve as a biomarker for iron status in RUG, especially RUG with lower fat mass.

## 1. Introduction

Iron deficiency can have severe effects on athletes’ performance and health [1,2]. Special attention to this condition is paid in lean athletes, who are at potential risk of exercise-induced anemia [3,4]. To address the issue of exercise-induced anemia, a biomarker of iron status is necessary to predict the condition.

Hepcidin has been recently reported as a surrogate iron status marker in athletes [5,6,7,8]. Hepcidin degrades ferroprotein export channels on the surfaces of macrophages; thus, high hepcidin levels reduce iron recycling and absorption from the intestine [9,10]. However, normal hepcidin levels among athletes remain a topic of debate [11,12]; while one study shows that the levels are higher in athletes [11], another reports that the levels are not elevated [12]. This discrepancy is partly ascribed to the fact that lean athletes have low ferritin levels, and resting hepcidin levels are generally regulated by the ferritin level [13,14,15,16]; however, in some cases, the hepcidin level is independent of the ferritin level. For instance, acute exercise increases the hepcidin level without changing the ferritin level due to exercise-induced inflammation caused by elevated interleukin-6 (IL-6) levels [14,15,16]. This higher hepcidin level can remain sustained for 2–3 days while the IL-6 level is in the normal range soon after the exercise [16,17,18]. In addition, the hepcidin level is higher during an intensified training period than during a regular training period, independent of the ferritin level [19].

In contrast, recent studies have postulated that lower energy availability increases hepcidin levels [8,20,21]. This led us to hypothesize that an athlete with a lower body mass index (BMI) has a higher hepcidin level, independent of the ferritin level. To test this hypothesis, we recruited rugby football players. Rugby football is a unique sport in which athletes have a range of body weights and spend a large amount of energy on the field (40 min × 2 times). The differing degrees of body fat encountered in rugby players [20,21,22,23] may provide a deeper understanding of the effects of regular exercise on hepcidin levels. Usually, body fat is positively correlated with the hepcidin level [13,24,25,26,27]. Interestingly, however, recent accumulative studies [16,17,18] have suggested that lower energy availability could increase hepcidin levels. Therefore, we aimed to analyze whether rugby players with lower body fat had higher levels of hepcidin and/or hepcidin/ferritin ratios. Further, we also investigated the relationship between the hepcidin level and body fat in male university rugby players, considering the ferritin level.

## 2. Materials and Methods

We enrolled 40 male university rugby football players in this study (RUG group). All the players were members of the A-League, the highest level of the local league in Japan. They practiced for 3 h a day–6 days a week. We also recruited 40 non-athlete controls (CON group) from the same university by displaying posters across the campus. The subjects in the CON group did not engage in regular exercise (Table 1). This study was conducted in accordance with the principles of the Declaration of Helsinki (Fortaleza, Brazil, 2013). The protocol was approved by the institutional review boards of Kyoto Medical Center (No. 15–089) and was registered at the University Hospital Medical Information Network (UMIN Clinical Trials Registry) center (UMIN000020236). Written informed consent was obtained from all participants. All measurements were collected between 11:00 and 13:00, 24 h after exercise.

Bodyweight, body fat content, and skeletal muscle mass were measured using an Inbody 430 analyzer (Biospace, Seoul, Korea). The body fat and skeletal muscle were estimated using the impedance method by measuring the voltage drop while passing a current between electrodes [13,22,23].

Peripheral blood tests were performed to determine red blood cell counts, hemoglobin and hematocrit levels, mean corpuscular volume, mean corpuscular hemoglobin, and mean corpuscular hemoglobin concentration. To quantify the active form of hepcidin, namely hepcidin-25, serum samples were mixed with synthetic human hepcidin (Peptide Institute, Osaka, Japan) as an internal standard and passed through a reverse-phase PLRP-S column (5 mm, 300 Å, 150 × 3 × 2.1 mm; Varian, Inc., Palo Alto, CA, USA). The levels of hepcidin-25 in the eluate were measured using a 4000 QTRAP liquid chromatography–tandem mass spectrometry system (Applied Biosystems, Foster City, CA, USA) [13]. A LABOSPECT 008α automatic colorimetric analyzer (Hitachi High-Tech America, Inc., Schaumburg, IL, USA) was used to measure serum levels of iron. To test the levels of stored iron, we measured the concentrations of serum ferritin using an iatro ferritin kit (LSI Medience Corporation, Tokyo, Japan). The IL-6 levels were quantified using an enzyme-linked immunosorbent assay kit (EMD Millipore Corporation, Billerica, MA, USA) to confirm whether the exercise-induced inflammation marker IL-6 level was normal in athletes and also to confirm the absence of exercise-induced inflammation in the subjects in the RUG group.

Data were expressed as means ± standard error. The differences between the groups were analyzed using the independent Student’s *t*-test. To compare the three groups, we used a one-way ANOVA with Tukey’s post hoc test. The correlation was assessed (linear and exponential trendlines) using Pearson’s correlation coefficient. Prevalence of iron deficiency anemia (ferritin < 30 ng/mL) was compared using the Chi-square (χ^2^) test. A *p*-value < 0.05 was considered statistically significant. All statistical analyses were performed using IBM SPSS Statistics software for Windows, version 22.0 (IBM Corp., Armonk, NY, USA) or GraphPad Prism 8 (GraphPad Software, Inc., La Jolla, CA, USA).

## 3. Results

The anthropometric characteristics of the study participants are shown in Table 1. The participants in the RUG and CON groups had an average age of 20.1 ± 0.1 years and 20.9 ± 0.3 years, respectively. The RUG group had significantly greater body weight (BMI), lean mass, and body fat than did the CON group (*p* < 0.05).

The iron metabolism-related parameters are shown in Table 2. The RUG group had lower ferritin levels than the CON group (67.6 ± 5.7 vs. 97.3 ± 5.7 ng/mL; *p* < 0.05). Iron deficiency anemia (ferritin < 30 ng/mL) was observed in 3 of the 40 (10.0%) subjects in the RUG group and in none of the CON subjects (RUG vs. CON, *p* < 0.05). Although the ferritin levels differed between the groups, there was no difference in the level of hepcidin-25. As numerous previous studies have suggested that ferritin may be the most important factor of hepcidin expression [13,14,15], we decided to investigate the relationship between ferritin and hepcidin-25 levels in male university rugby football players compared to controls. Interestingly, the ratio of hepcidin-25/ferritin was 61.9% greater in the RUG group than in the CON group (0.34 ± 0.24 vs. 0.21 ± 0.18; *p* < 0.05). The Il-6 level in the RUG group was 1.2 ± 0.1, which is considered within the normal reference range.

Figure 1A shows the relationship between hepcidin-25 and ferritin in the RUG and CON groups. We observed a significant relation both in RUG (*r* = 0.47, *p* < 0.05) and CON (*r* = 0.54, *p* < 0.05). Since IL-6 is a key factor behind acute exercise-induced inflammation, we confirmed that there was no significant relationship between IL-6 and ferritin (Figure 1B, *r* = 0.23, *p* = 0.16) or hepcidin (Figure 1C, *r* = 0.19, *p* = 0.23) in the RUG group.

Figure 2 shows the relationship between hepcidin-25 and body composition in the RUG and CON groups. Interestingly, a U-shaped curve was observed between hepcidin-25 and body fat (Figure 2A), fat mass index (Figure 2B), body fat percentage (Figure 2C), and BMI (Figure 2D) in both groups. However, this relationship was not observed between hepcidin-25 and lean mass (Figure 2E). The correlations between the level of hepcidin-25 and the amount of body fat (*r* = 0.79 vs. *r* = 0.42; *p* < 0.05), fat mass index (*r* = 0.79 vs. *r* = 0.45, *p* < 0.05), body fat percentage (*r* = 0.77 vs. *r* = 0.41; *p* < 0.05), and BMI (*r* = 0.39 vs. *r* = 0.69; *p* = 0.15) were either significant or tended toward significance in the RUG group but not in the CON group. Our data suggest that athletes who either had a higher body fat mass or lower body fat mass had higher hepcidin-25 levels. To estimate the optimal range of body fat that sustains hepcidin level, we substituted the average hepcidin-25 level (21.3 ng/mL) into the formula of a quadratic function. As a result, the body fat (16.2–24.2 kg), fat mass index (3.1–10.3 kg/m^2^), %body fat (12.6–29.4%), and BMI (22.9–33.8 kg/m^2^) were suggested as optimal ranges in the RUG group. It is known that obesity increases hepcidin levels, which may be due to inflammation [22,23,24]. However, a lower body fat mass group with a higher hepcidin-25 level in athlete is new information.

To obtain further information that the hepcidin-25 levels were increased in athletes with lower body fat, we divided the RUG group into three groups; the players who had a fat mass index <3.1 kg/m^2^, 3.1–10.3 kg/m^2^, and >10.3 kgm^2^, which corresponded to the average hepcidin-25 value (21.3 ng/mL) of the quadratic function expression in Figure 2B. Notably, the lowest fat mass index group had—(1) higher hepcidin-25 levels than the middle fat mass index group; (2) lower ferritin levels compared to the highest body fat index group; and (3) higher hepcidin-25/ferritin ratios than the highest and middle-fat mass index groups (Figure 2A–C). Interestingly, the Hb level in the lowest fat mass index group tended to be lower than that of the highest fat mass index group (16.3 ± 0.7 vs. 15.3 ± 0.6, *p* = 0.06 via AVOVA, *p* < 0.05 via *t*-test) (Figure 3). The numbers of iron-deficient athletes was >10.3 (0/10 athletes, 0%), 3.3–10.3 (2/24 athletes, 8.3%), and <3.1 (1/6 athletes, 16.6%). The data did not differ significantly among the groups. The value of IL-6 for each fat mass index group were >10.3 (1.42 ± 0.18), 3.3–10.3 (1.11 ± 0.11), and <3.1 (1.17 ± 0.42). There was no significant difference between the three groups (one-way ANOVA analysis, *p* = 0.13).

## 4. Discussion

The main findings of the present study were as follows: (1) the relationship between the level of hepcidin-25 and the degree of body fat was U-shaped, (2) the hepcidin-25/ferritin ratio was relatively elevated in the RUG group than in the CON group, and (3) lean rugby football players had higher hepcidin-25/ferritin ratio compared to rugby football players who had normal or high body fatness. Furthermore, the correlation coefficients between hepcidin-25 and body fatness were greater in the RUG group than in the CON group. These results suggest that hepcidin-25 level increased in the lean as well as obese rugby football players independent of the ferritin level.

Recently, many researchers have attempted to find a biomarker for exercise-induced anemia, and the hepcidin level was suggested as a potential biomarker for lower energy availability [8,13,16,17]. It is well known that the hepcidin level is positively correlated with the ferritin level [13,14,15,16]. In the present study, we confirmed this relationship in male university rugby players. However, we found a steeper slope in the RUG group than in the CON group. Moreover, the hepcidin/ferritin ratio after 24 h exercise was significantly higher in the RUG group than in the CON group. Physiologically, elevated hepcidin levels have been found to decrease ferritin levels, which, in turn, regulate hepcidin levels [8,24]. Therefore, we speculated that the hepcidin levels would be relatively elevated in the RUG group of our study. Vigorous exercise has been found to cause chronic inflammation and to thereby increase hepcidin levels through IL-6 [8]. However, the IL-6 levels were within the normal range in the present study. A previous study showed that 3 days of reduced energy intake increased exercise-induced hepcidin levels in athletes compared with a normal carbohydrate diet for an equivalent duration [15]. In the present study, as shown in Figure 2A–C, the hepcidin-25/ferritin ratio is increased in lean rugby football players compared to rugby football players who have a normal or higher body fatness. Taken together with the findings from previous studies [8,13,16,17], our data suggest that the level of hepcidin-25 increased in both obese and lean subjects, albeit more so in male rugby football players. The hepcidin-25 levels followed U-shaped curves that increase beyond the normal range (fat mass index (3.1–10.3 kg/m^2^) or %body fat (12.6–29.4%)) in either direction. Together with previous studies, the present study shows that the hepcidin-25 level/ferritin ratio is a potential biomarker to detect exercise-induced anemia in lean athletes. Indeed, our data indirectly show that the lower fatness group has lower ferritin and hemoglobin concentration with higher hepcidin-25/ferritin ratio.

Our study has several limitations. First, the sample size was relatively small and was drawn from a single rugby football club. Second, as this was a simple observational study, the causality of the results, especially the U-shaped curve between body fat and hepcidin, remains unclear. However, we speculated that this U-shaped relationship is related to energy availability. Future studies should consider evaluating the relationship between energy intake and the various body positions while playing rugby against the backdrop of the mechanisms of hepcidin-25 elevation presented in this report. Further, dietary intervention is warranted to alter the degree of body fat in rugby football players. Besides the dietary intake, future studies should also consider investigating the relationship between other inflammation markers and hepcidin-25. Finally, we collected the blood samples after 24 h routine practice. However, previous studies have reported elevated hepcidin levels for up to 3 days [16,17,28]. Therefore, future studies should consider evaluating blood samples collected on different days.

## 5. Conclusions

Hepcidin-25 was elevated in relation/ferritin ratio in male university rugby football players when compared to non-athlete subjects. Moreover, the relationship between the level of hepcidin-25 and body fat followed a U-shaped curve in male university rugby football players. For rugby football players, obesity as well as being too lean can be a risk for increasing hepcidin-25 levels. Importantly, the lower body fatness group has lower hemoglobin, ferritin, and hepcidin-25/ferritin ratio. The collective results indicate that a hepcidin 25 level or hepcidin-25/ferritin ratio may serve as a biomarker for iron status in lean rugby football players. However, we were unable to obtain direct evidence that hepcidin-25/ferritin ratio is a biomarker for iron status. Further studies including large numbers of participants are needed to provide evidence that hepcidin-25/ferritin ratio is a biomarker for iron status.

## Figures and Tables

**Figure 1 nutrients-13-02993-f001:**
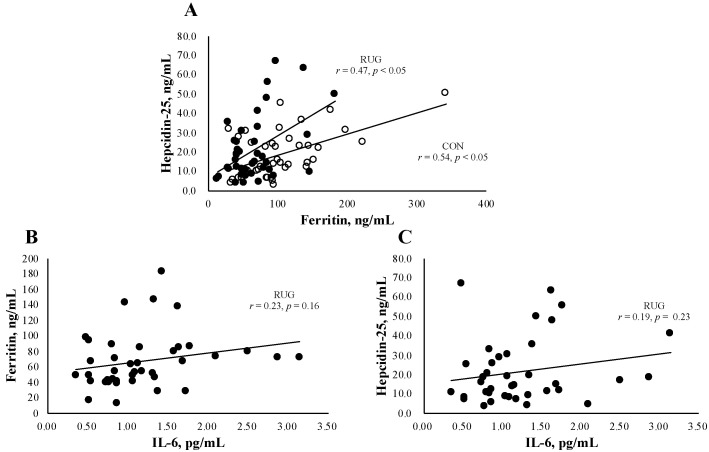
Relationship between hepcidin-25 and ferritin or IL-6. The correlation analyses were conducted between hepcidin-25 and ferritin (**A**), hepcidin-25 and IL-6 in the RUG group (**B**), and ferritin and IL-6 in the RUG group (**C**). ●RUG group, ◯CON group.

**Figure 2 nutrients-13-02993-f002:**
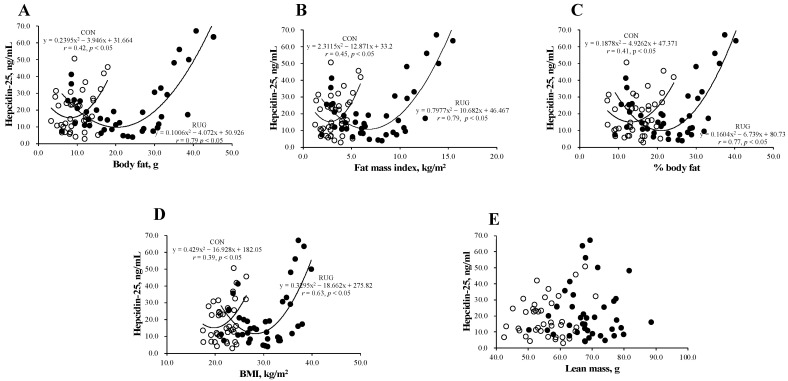
Relationship between hepcidin-25 and anthropometric parameters in rugby football players and non-athlete controls. The correlation analyses were conducted between hepcidin-25 and (**A**) the amount of body fat, (**B**) fat mass index, (**C**) the body fat percentage, (**D**) BMI, and (**E**) lean mass. The correlation coefficients between hepcidin-25 and (**A**) the amount of body fat (*r* = 0.79 vs. *r* = 0.42; *p* < 0.05), (**B**) fat mass index (*r* = 0.79 vs. *r* = 0.45, *p* < 0.05), and (**C**) body fat percentage (*r* = 0.77 vs. *r* = 0.41; *p* < 0.05) were significantly greater in the RUG group than in the CON group. RUG, rugby football players; CON, non-athlete controls; BMI, body mass index. ●RUG group, ◯CON group.

**Figure 3 nutrients-13-02993-f003:**
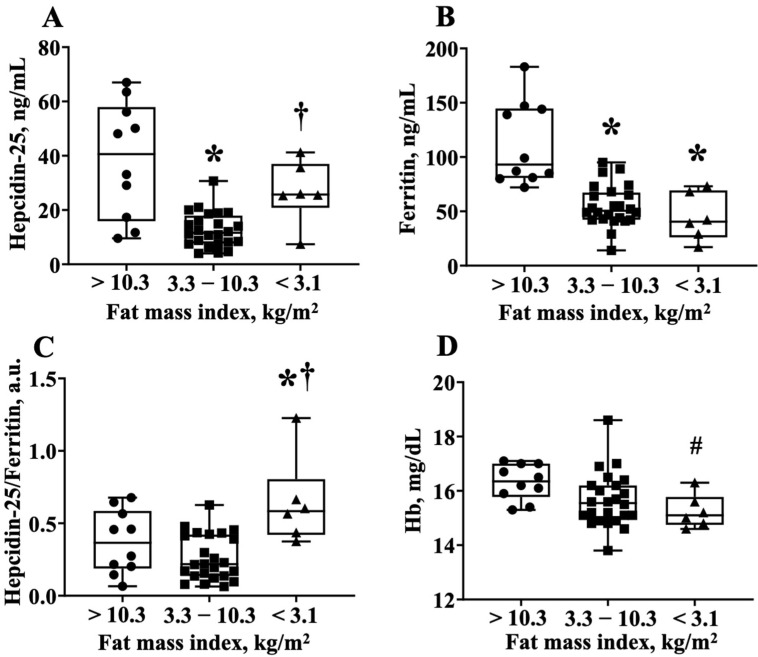
The hepcidin-25/ferritin ratio is higher in the lower body fat group of the rugby football players. (**A**) Hepcidin-25 levels, (**B**) ferritin levels, (**C**) hepcidin-25/ferritin ratio, and (**D**) Hb levels are compared among the three groups. These groups are divided by fat mass index <3.1 kg/m^2^, 3.1–10.3 kg/m^2^, and >10.3 kgm^2^, which corresponds to the average hepcidin-25 value (21.6 ng/mL) to the regression equation of Figure 2B. Hb, hemoglobin. * *p* < 0.05, vs. fat index >10.3 kg/m^2^, † *p* < 0.05, vs. fat index 3.3–10.3 kg/m^2^, # *p* = 0.06, vs. fat index >10.3 kg/m^2^.

**Table 1 nutrients-13-02993-t001:** Comparison of anthropometric characteristics between rugby players and sedentary controls.

Characteristics	CON (*n* = 40)	RUG (*n* = 40)	*p* Value
Height, cm	172.5 ± 0.8	174.1 ± 0.9	0.19
Body weight, kg	65.3 ± 1.4	90.9 ± 2.5	<0.05
BMI, kg/m^2^	21.8 ± 0.3	30.0 ± 0.8	<0.05
Body fat, kg	9.8 ± 0.6	23.3 ± 1.3	<0.05
Body fat, %	14.8 ± 0.7	22.1 ± 1.7	<0.05
Lean mass, kg	55.4 ± 1.0	68.8 ± 1.2	<0.05

RUG, rugby football players group; CON, non-athlete control group; BMI, body mass index.

**Table 2 nutrients-13-02993-t002:** Comparison of iron metabolism related parameters between rugby players and non-athlete controls.

Parameters	CON (*n* = 40)	RUG (*n* = 40)	Normal Reference	*p* Value
Hepcidin-25, ng/mL	19.1 ± 1.9	21.3 ± 2.6	-	0.50
Ferritin, ng/mL	97.3 ± 6.7	67.6 ± 5.7	9.0–275	<0.05
%ferritin > 30 ng/mL	0	10	-	<0.05
Hepcidin-25/Ferritin ratio	0.21 ± 0.18	0.34 ± 0.24	-	<0.05
Fe, mg/dL	124.3 ± 6.9	118.1 ± 38.2	50–200	0.50
RBCs × 10^4^/mL	509.3 ± 5.8	521.0 ± 29.8	430–570	0.12
Hb, g/dL	15.4 ± 0.2	15.8 ± 0.1	13.5–17.5	0.19
Hct, %	46.1 ± 0.4	46.9 ± 2.6	39.7–52.4	0.17
MCV, fL	90.8 ± 0.6	90.2 ± 3.8	85–102	0.51
MCH, pg	30.2 ± 0.18	30.2 ± 1.2	28.0–34.0	0.68
MCHC, %	33.4 ± 0.2	33.5 ± 0.8	30.2–35.1	0.83
IL-6, pg/mL	-	1.2 ± 0.1	<7.0	-

RUG, rugby football players group; CON, non-athlete control group; RBC, red blood cell; Hb, hemoglobin; Hct, hematocrit; MCV, mean corpuscular volume; MCH, mean corpuscular hemoglobin; MCHC, mean corpuscular hemoglobin concentration; IL-6, interleukin-6.

## Data Availability

The datasets generated during and/or analyzed during the current study are not publicly available but are available from the corresponding author on reasonable request.

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
