# Peer review of "The Hepcidin-25/Ferritin Ratio Is Increased in University Rugby Players with Lower Fat Mass"

_nutrients, 2021, doi:10.3390/nu13092993_

Round 1

Reviewer 1 Report

This is an interesting study examining the relationship between hepcidin25, ferritin and various anthropometric measurements in male rugby players. We are still developing our knowledge of iron metabolism in athletes and rugby players are interesting in that their sport demands both aerobic and anaerobic fitness and athletes have differing body compositions.

My two major concerns with this study is that the actual relationship between ferritin and hepcidin are not shown - and I believe this needs to be shown in graphical form, and also although the authors stated they measured IL6 I don't see the data for this - and this data needs to be included in the analysis of the relationship between hepcidin, ferritin and ideally haemoglobin.

The big confounder when measuring either hepcidin or ferritin in the context of assessing body iron status is inflammation - both are acute phase reactants that increase with inflammation even in the face of iron deficiency when we would predict that they should be low - and the results of this study can't be meaningfully understood without knowing the inflammation status of the subjects.

Other points

line 21- remove "the" before hepcidin

line 25 - rather than slightly and higher please be specific - how much higher or lower using percentages

line 32 - lean body MASS?

line 38 - why is a biomarker urgently required in particular for this group? please explain and provide evidence

lines 49-50 - please explain why hepcidin increase is independent of ferritin - inflammation raises ferritin as well so why the disconnect in this context? Please provide evidence to support

line 60 - why would the authors predict lower body fat would lead to higher hepcidin? I would predict that lower body fat leads to lower iron status and hence LOWER hepcidin. Please support with research evidence in people

Table 2 - please provide the normal reference ranges for all parameters

also - do the authors mean 10% had ferritin <30 ng/ml? - they have written >

Lastly, where is IL6 for controls?

lines 130 131 - please specify how much higher and lower

Please provide a graph plotting hepcidin, ferritin and IL6 to validate your measurement model. Perhaps also include haemoglobin?

line 151- please check spelling

line 152- how much lower was haemoglobin? physiologically relevant?

Can the authors provide a suggested physiological mechanism to explain their counter-intuitive findings? why would hepcidin be higher with lower ferritin?

Why is there a U shaped curve between body fat and hepcidin?

These findings do not all make sense to me, and I would like to know the authors understanding of possible physiological emchamnism to explain - and as they measured IL6 if it is inflammation they should demonstrate this 

Author Response

We wish to re-submit the attached manuscript for reconsideration. The manuscript ID is nutrients-1319201.

The manuscript has been rechecked and appropriate changes have been made in accordance with the reviewers’ suggestions. The responses to their comments have been prepared and given below.

We thank you and the reviewers for your thoughtful suggestions and insights, which helped us enrich the manuscript and produce a better and more balanced account of the research. We hope that the revised manuscript is now suitable for publication in your journal.

We look forward to your response.

Sincerely,

Naoki Sakane, MD, PhD 

Q1. My two major concerns with this study is that the actual relationship between ferritin and hepcidin are not shown - and I believe this needs to be shown in graphical form, and also although the authors stated they measured IL6 I don't see the data for this - and this data needs to be included in the analysis of the relationship between hepcidin, ferritin and ideally haemoglobin.

A1. I have added the new figure illustrating the relationship between Ferritin vs. Hepcidin as Fig 1 and have also described the same in the following sentences (lines 150–154 and 157–159). The relationship between hemoglobin and hepcidin-25 was not significant for both RUG (r = 0.21, P = 0.19) and CON groups (r = -0.11, P = 0.56) and was therefore considered not important.

Q2. The big confounder when measuring either hepcidin or ferritin in the context of assessing body iron status is inflammation - both are acute phase reactants that increase with inflammation even in the face of iron deficiency when we would predict that they should be low - and the results of this study can't be meaningfully understood without knowing the inflammation status of the subjects.

A2. We know that hepcidin levels are increased by the IL-6/Stat3 pathway (Kitamura et al. Cancer Sci 2017). However, we did not observe an increase in IL-6 levels among athletes in our study. The same has been clarified in relevant sentences in the manuscript (Lines 94, 147, 197 and 229).

Other points

Q3. line 21- remove "the" before hepcidin

A3. We have removed the "the" before hepcidin (line 21).

Q4. line 25 - rather than slightly and higher please be specific - how much higher or lower using percentages

A4. We have added the % value for these values (lines 25 and 26).

Q5. line 32 - lean body MASS?

A5. We have added mass and the title (line 32 and Title).

Q6. line 38 - why is a biomarker urgently required in particular for this group? please explain and provide evidence

A6. We have justified the reason for saying ‘to predict the condition before exercise-induced anemia’ (line 39).

Q7. lines 49-50 - please explain why hepcidin increase is independent of ferritin - inflammation raises ferritin as well so why the disconnect in this context? Please provide evidence to support

A7. We have changed this sentence as below (lines 50-52).

For instance, acute exercise increases the hepcidin level without changing the ferritin level due to exercise-induced inflammation caused by elevated IL-6 levels [14-16]. This higher hepcidin level can remain sustained for 2–3 days while the IL-6 level is the normal range soon after the exercise [16,17].

Q8. line 60 - why would the authors predict lower body fat would lead to higher hepcidin? I would predict that lower body fat leads to lower iron status and hence LOWER hepcidin. Please support with research evidence in people

A8. We have added the relevant sentence as below.

Interestingly, however, recent accumulative studies [16-18] have suggested that lower energy availability could increase hepcidin levels. Therefore, we aimed to analyze whether rugby players with lower body fat had higher levels of hepcidin and/or hepcidin/ferritin ratios (Line 62-65).

Q9. Table 2 - please provide the normal reference ranges for all parameters also - do the authors mean 10% had ferritin <30 ng/ml? - they have written

A9. We have added the normal reference value in Table 2. This (Iron deficiency anemia; ferritin < 30 ng/mL) 10% value is consistent with a previous study (Smith et al. J Sports Med Phys Fitness 2020)

Q10. Lastly, where is IL6 for controls?

A10. We know that Il-6 is increased by acute exercise. Therefore, we do not anticipate increased IL-6 levels in CON (healthy lean non-exercised university students) group. The IL-6 test kit costs approximately $2,000–$3,000. In this study, the Il-6 was evaluated in rugby football players to confirm whether their Il-6 is not higher than normal reference. To acknowledge this point, we have added the following sentences to the manuscript.

Method (line 94): The IL-6 levels were quantified using an enzyme-linked immunosorbent assay kit (EMD Millipore Corporation, Billerica, MA, USA) to confirm whether the exercise-induced inflammation marker IL-6 level was normal in athletes

Result (line 147): The Il-6 level in the RUG group was 1.2±0.1, which is considered within the normal reference range.

.

Q11. lines 130 131 - please specify how much higher and lower

A11. We have added the % of the value (line 146).

Q12. Please provide a graph plotting hepcidin, ferritin and IL6 to validate your measurement model. Perhaps also include haemoglobin?

A12. We have added the graph between hepcidin vs. ferritin, ferritin vs. Il-6, and hepcidin vs. IL6 in Fig 1. The relationship between hemoglobin and hepcidin-25 was found to be not significant for both RUG (r = 0.21, P = 0.19) and CON group (r = -0.11, P = 0.56).

Q13. line 151- please check spelling.

A13. We have had our manuscript proofread by a native English-speaking editor.

Q14. line 152- how much lower was haemoglobin? physiologically relevant?

A14. We have added normal reference to the Table, as pointed out in the Q9.

Q15. Can the authors provide a suggested physiological mechanism to explain their counter-intuitive findings? why would hepcidin be higher with lower ferritin?

A15. This is a critical question that needs to be addressed. It might be that low energy availability induced-repeated inflammation increases hepcidin level. However, since this is a cross-sectional study, we do not have concrete evidence to stake claims in this aspect. Further studies are needed to elucidate the mechanism. We have added this statement to the discussion section (Lines 243-248).

Second, as this was a simple observational study, the causality of the results, especially the U-shaped curve between body fat and hepcidin, remains unclear. However, we speculated that this U-shaped relationship is related to energy availability. Future studies should consider evaluating the relationship between energy intake and the various body positions while playing rugby against the backdrop of the mechanisms of hepcidin-25 elevation presented in this report. Further, dietary intervention is warranted to alter the degree of body fat in rugby football players. Besides the dietary intake, future studies should also consider investigating the relationship between other inflammation markers and hepcidin-25.

Q16. Why is there a U shaped curve between body fat and hepcidin?

A16. We believe that the response to this question is covered within the Q15.

Reviewer 2 Report

In the study by Nirengi, S et al, authors aim to investigate the relationship between hepcidin and body fat levels in rugby football players with the goal of identifying a biomarker for exercise-induced anemia. 40 males were recruited for each of the groups: university rugby football players (RUG) and non-athletes (CON). Authors hypothesize that lean athlete have higher hepcidin level, independent of ferritin level.  Anthropometric measurements and samples for haematologic and iron measurements were collected around the same time of day, 24 hours after exercise, presumably to measure basal levels in both groups. Hepcidin-25 were plotted against anthropometric parameters, authors observed a U-shaped curve between hepcidin-25 and body fat. Authors also report in lean athletes, an elevated hepcidin-25/ferritin ratio. Authors propose hepcidin-25 or a hepcidin-25/ferritin ratio as a potential biomarker for iron metabolism.

In general, the finding that low fat mass index athletes have increased hepcidin is novel, most studies have compared normal-weight individuals compared to overweight/obese individuals, albeit they do not look specifically at athletes (Stoffel NU et al, IJO 2020). However, I question the importance of the hepcidin-25/ferritin ratio in providing insight into exercise induced anemia in athletes. Although the athletes varied in their fat mass index, their iron metabolism and hematological parameters were all normal. The study is very much descriptive.

  1. Authors state that ‘hepcidin levels in athletes are controversial’ (line 43), could you please elaborate or clarify what is meant by controversial. If one study showed higher levels in athletes (line 44), do you mean to say that hepcidin levels are typically lower in athletes?
  2. The finding that in low fat mass index athletes have increased hepcidin is interesting. On average IL-6 is low in the RUG group, it would be interesting to either correlate IL-6 with hepcidin levels in both groups or look at IL-6 levels in the >10.3, 3.3-10.3, <3.1 groups.
  3. Samples in the current study were collected 24hrs post-exercise, were any samples collected more than 24hrs post-exercise? Since increased levels of hepcidin are reported to persist for up to 72hrs post exercise (Ziemann E, Sports Sci Med. 2013).
  4. Line 20: please clarify what you mean by “hepcidin metabolism”, do you mean changes in hepcidin concentrations?
  5. Line 41: to my knowledge, hepcidin is the only known cytokine to induce ferroportin degradation. If indeed adipokine and hepatokine degrade ferroportin, please include the appropriate citations.
  6. Please provide IL-6 measurements for the CON group, increased baseline inflammation in the CON group could account for the observed higher ferritin levels.
  7. Please explain the discrepancy in values for Ferritin between Table 2 and in the text (line 113).

Author Response

We wish to re-submit the attached manuscript for reconsideration. The manuscript ID is nutrients-1319201.

The manuscript has been rechecked and appropriate changes have been made in accordance with the reviewers’ suggestions. The responses to their comments have been prepared and given below.

We thank you and the reviewers for your thoughtful suggestions and insights, which helped us enrich the manuscript and produce a better and more balanced account of the research. We hope that the revised manuscript is now suitable for publication in your journal.

We look forward to your response.

Sincerely,

Naoki Sakane, MD, PhD

In general, the finding that low fat mass index athletes have increased hepcidin is novel, most studies have compared normal-weight individuals compared to overweight/obese individuals, albeit they do not look specifically at athletes (Stoffel NU et al, IJO 2020). However, I question the importance of the hepcidin-25/ferritin ratio in providing insight into exercise induced anemia in athletes. Although the athletes varied in their fat mass index, their iron metabolism and hematological parameters were all normal. The study is very much descriptive.

Q1. Authors state that ‘hepcidin levels in athletes are controversial’ (line 43), could you please elaborate or clarify what is meant by controversial. If one study showed higher levels in athletes (line 44), do you mean to say that hepcidin levels are typically lower in athletes?

A1. We have changed this sentence as below (Line 44-46).

However, hepcidin levels in athletes are controversial [11, 12], with one study showing the levels to be higher in athletes [11] while the other one is not elevated [12].

Q2. The finding that in low fat mass index athletes have increased hepcidin is interesting. On average IL-6 is low in the RUG group, it would be interesting to either correlate IL-6 with hepcidin levels in both groups or look at IL-6 levels in the >10.3, 3.3-10.3, <3.1 groups.

A2. We have added the Il-6 levels for these groups (lines 197-199) and have illustrated the comparison of IL-6 vs. hepcidin in Fig 1.

Q3. Samples in the current study were collected 24hrs post-exercise, were any samples collected more than 24hrs post-exercise? Since increased levels of hepcidin are reported to persist for up to 72hrs post exercise (Ziemann E, Sports Sci Med. 2013).

A3. We collected the blood sample after 24hrs. We have added relevant information regarding time course in the limitation section as below (Line 251-253).

Finally, we collected the blood samples after 24h routine practice. However, previous studies have reported elevated hepcidin levels for up to 3 days [16, 17, 28]. Therefore, future studies should consider evaluating blood samples collected on different days.

Q4. Line 20: please clarify what you mean by “hepcidin metabolism”, do you mean changes in hepcidin concentrations?

A4. We have changed metabolism to concentration (line 20).

Q5. Line 41: to my knowledge, hepcidin is the only known cytokine to induce ferroportin degradation. If indeed adipokine and hepatokine degrade ferroportin, please include the appropriate citations.

A5. We have wanted to say that hepcidin is either hepatokine or adipokine. We have rephrased the sentence to include this information as below (lines 42-44).

Hepcidin, either adipokine or hepatokine, degrade ferroportin export channels on the surfaces of macrophages; thus, high hepcidin levels reduce iron recycling and absorption from the intestine [9,10].

Q6. Please provide IL-6 measurements for the CON group, increased baseline inflammation in the CON group could account for the observed higher ferritin levels.

A6. We know that Il-6 is increased by acute exercise. Therefore, we do not anticipate increased IL-6 levels in CON (healthy lean non-exercised university students) group. The IL-6 test kit costs approximately $2,000–$3,000. In this study, the Il-6 was evaluated in rugby football players to confirm whether their Il-6 is not higher than normal reference. To acknowledge this point, we have added the following sentences to the manuscript.

Method (line 94): The IL-6 levels were quantified using an enzyme-linked immunosorbent assay kit (EMD Millipore Corporation, Billerica, MA, USA) to confirm whether the exercise-induced inflammation marker IL-6 level was normal in athletes

Result (line 147): The Il-6 level in the RUG group was 1.2±0.1, which is considered within the normal reference range.

Q7. Please explain the discrepancy in values for Ferritin between Table 2 and in the text (line 113).

A7. We sincerely regret this discrepancy. We have corrected it in the modified version of the manuscript.

Round 2

Reviewer 1 Report

Thank you for the opportunity to read the updated submission - the authors have clearly worked hard and the manuscript is improved.

There are still things that need to be corrected/clarified

If I am interpreting the reply correctly IL-6 was not measured in controls for financial reasons- this is a serious limitation and needs to be acknowledged as such.

Following are issues/suggested corrections

line 25 - check grammar

line 31-32 - enhancing for what?

line 39-40 - award wording

line 42- what does adipokine or hepatokine mean? hepcidin is part of the innate immune system and the master iron regulator

Hepcidin is not a surrogate for iron metabolism - it is part of the control mechanism of iron metabolism - do they mean a surrogate iron status marker?

line 57 - check grammar

all the figures that have the linear regression derived equation of the line - y= 12.96x + 52 - for example- need to have the equation removed from the graph - please review other scientific articles to see how to present this data

line 154-155 - "could be? 

line 173 - AVOVA?

line 190 - correlation coefficients showing correlation of what two factors?

Lastly but most significantly - the authors' conclusions that the hepcidin-ferritin ratio could be used as a biomarker to assess risk for exercise-associated anaemia - is not supported by their discussion or data. How could this ratio be a biomarker for risk of anaemia? on what basis? the ratio has a u shaped relationship with body fat mass - how does this relate to risk for anaemia? A much more compelling/cogent argument has to be made  to support this conclusion. I just don't see the connections.

Only three subjects in this study had iron deficiency - although very limited this is where I would start as a hypothesis generating discussion

Author Response

Thank you for your comments. They were all meaningful and helpful to improve our manuscript. Our revisions are indicated using red font.

Q1. line 25 and 57 - check grammar

A1. The revision was done with the assistance of a native English speaking science editor.

Q2. line 31-32 - enhancing for what?

A2. We have deleted this sentence on line 31.

Q3. line 39-40 - award wording

A3. We have changed this sentence. (lines 38-39)

Q4. line 42- what does adipokine or hepatokine mean? hepcidin is part of the innate immune system and the master iron regulator

A4. You are correct. We appreciate your noting our mistake. We have deleted adipokine or hepatokine. (line 41)

Q5. Hepcidin is not a surrogate for iron metabolism - it is part of the control mechanism of iron metabolism - do they mean a surrogate iron status marker?

A5. We have changed it to ‘iron status marker’. (line 19, 32, 34, 38, 40 and 250)

Q7. all the figures that have the linear regression derived equation of the line - y= 12.96x + 52 - for example- need to have the equation removed from the graph - please review other scientific articles to see how to present this data

A7. We have deleted the equation from Fig. 1. However, for Fig. 2, estimated the optimal body fat using this equation (see lines 150-151 and 164-166). Therefore, the equation was retained.

Q8. line 154-155 - "could be? 

A8. We changed ‘could be’ to ‘is’. (line 167-168)

As previously commented on by Reviewer 2, the lower body fat mass group of athletes is now provided as the correct new information.

Q9. line 173 - AVOVA?

A9. We have changed it to ‘one-way ANOVA analysis’. (line 190)

Q10. line 190 - correlation coefficients showing correlation of what two factors?

A10. We have added information on the correlation between hepcidin-25 and body fatness on (line 205).

Q11. Lastly but most significantly - the authors' conclusions that the hepcidin-ferritin ratio could be used as a biomarker to assess risk for exercise-associated anaemia - is not supported by their discussion or data. How could this ratio be a biomarker for risk of anaemia? on what basis? the ratio has a u shaped relationship with body fat mass - how does this relate to risk for anaemia? A much more compelling/cogent argument has to be made to support this conclusion. I just don't see the connections. Only three subjects in this study had iron deficiency - although very limited this is where I would start as a hypothesis generating discussion

A11. Thank you for critical comment. Based on your suggestion we compared iron deficiency among the three groups. However, the number of iron deficient athletes was not statistically significant: 1/6 (16.7%) in 3.1 kg/m2 <, 2/24 (8.3%) in 3.1-10.3 kg/m2, and 0/10 (0%) in > 10.3 kgm2 groups.

Therefore, we have determined that our conclusion needed to be toned down. To do this, added a sentence regarding iron deficiency in the Results section (lines 186-188). We have also noted that more samples are necessary to conclude that the Hepcidin-25/ferritin ratio could be a biomarker of exercise-induced iron status (lines 230 and 250-256).

Reviewer 2 Report

Thank you for your revisions. I have no further comments.

Author Response

Thank you so much.